# Antibiotic Utilization during COVID-19: Are We Over-Prescribing?

**DOI:** 10.3390/antibiotics12020308

**Published:** 2023-02-02

**Authors:** Nataša Bednarčuk, Ana Golić Jelić, Svjetlana Stoisavljević Šatara, Nataša Stojaković, Vanda Marković Peković, Miloš P. Stojiljković, Nina Popović, Ranko Škrbić

**Affiliations:** 1Faculty of Medicine, University of Banja Luka, 78000 Banja Luka, The Republic of Srpska, Bosnia and Herzegovina; 2Department of Pharmacology, Toxicology and Clinical Pharmacology, Faculty of Medicine, University of Banja Luka, 78000 Banja Luka, The Republic of Srpska, Bosnia and Herzegovina; 3Department of Pharmacy, Faculty of Medicine, University of Banja Luka, 78000 Banja Luka, The Republic of Srpska, Bosnia and Herzegovina; 4Pharmacies Belladonna, 89101 Trebinje, The Republic of Srpska, Bosnia and Herzegovina; 5Academy of Sciences and Arts of the Republic of Srpska, 78000 Banja Luka, The Republic of Srpska, Bosnia and Herzegovina

**Keywords:** COVID-19, prescribing antibiotics, outpatient settings, antimicrobial resistance, medicine repurposing

## Abstract

The aims of this study were to analyze the utilization of antibiotics before (2018, 2019) and during the COVID-19 pandemic (2020) and the practice of prescribing antibiotics in outpatient settings for COVID-19 patients during the 2020–2022 period. The Anatomical Therapeutic Chemical Classification/Defined Daily Dose methodology was used for the analysis of outpatient antibiotic utilization in the Republic of Srpska. The data was expressed in DDD/1000 inhabitants/day. The rate of antibiotics prescribed to COVID-19 outpatients was analyzed using medical record data from 16,565 patients registered with B34.2, U07.1, and U07.2 World Health Organization International Classification of Diseases 10th revision codes. During 2020, outpatient antibiotic utilization increased by 53.80% compared to 2019. At least one antibiotic was prescribed for 91.04%, 83.05%, and 73.52% of COVID-19 outpatients during 2020, 2021, and the first half of 2022, respectively. On a monthly basis, at least one antibiotic was prescribed for more than 55% of COVID-19 outpatients. The three most commonly prescribed antibiotics were azithromycin, amoxicillin/clavulanic acid, and doxycycline. The trend of repurposing antibiotics for COVID-19 and other diseases treatment might be a double-edged sword. The long-term effect of this practice might be an increase in antimicrobial resistance and a loss of antibiotic effectiveness.

## 1. Introduction

The emergence of a new coronavirus that causes coronavirus disease 2019 (COVID-19) created a slew of issues for clinicians and researchers. The main problem since the beginning of the COVID-19 pandemic has been finding an appropriate and effective cure for the disease. In the absence of new therapeutics and due to the emergence of a large number of patients, clinicians started to repurpose approved or investigational drugs, especially those with antiviral and immunomodulatory effects [1]. Additionally, potential inappropriate use of antibiotics in COVID-19 patients appeared as a second big and long-term problem that might lead to an increase in antimicrobial resistance (AMR) and potentially take us to the post-antibiotic era [2,3,4]. Further, a study from United States showed that some commonly prescribed antibiotics, such as azithromycin, are very recalcitrant and pseudopersistant and can enter into aquatic ecosystems that receive the effluent from wastewater treatment plants [5]. In 2020, social distancing, a focus on isolation, and reductions in national and international travel could have reduced the spread of AMR. International travel has been consistently reported as an important risk factor for the acquisition of AMR pathogens [4]. In 2021 and 2022, all of the above have been facilitated, so AMR is mostly associated with inappropriate antibiotics prescribing. Appropriate prescription and optimized use of antimicrobials and antibiotics according to the principles of antimicrobial stewardship, along with quality diagnosis and aggressive infection control measures, may help prevent the occurrence of AMR during and after the pandemic [3]. Additionally, an English study conducted before COVID-19 showed that computerized decision support systems in hospitals had an influence on the antibiotic utilization by reducing their prescribing [6].

In all five versions of the World Health Organization (WHO) COVID-19 Clinical Management: Living Guidance, antibiotics were not recommended for the treatment of patients with suspected or confirmed mild and moderate COVID-19 unless there was a clinical suspicion or laboratory confirmation of bacterial infection [7,8,9]. In the first two versions of the WHO guidelines, empirical antibiotic therapy was recommended only for patients with severe COVID-19 [8,9]. Despite low rates of co-infections and secondary bacterial infections among hospitalized COVID-19 patients, data from various studies showed high rates of antibiotic prescribing [10,11,12,13]. Recent living rapid review and meta-analyses showed a high frequency of co-infections and secondary bacterial infections only in COVID-19 patients admitted to intensive care units [7]. Therefore, WHO now recommends that empirical antibiotic therapy should be considered only in those patients [7].

By our knowledge, only two studies, both from Italy, that assessed the practice of prescribing antibiotics for COVID-19 patients were conducted in outpatient settings. The first study, which included the severe acute respiratory syndrome coronavirus type 2 (SARS-CoV-2) positive patients treated in outpatient settings, showed that azithromycin and other antibiotics were prescribed, respectively, for 42.1% and 20.9% of patients [14]. A second study found that antibiotics were prescribed in outpatient settings for COVID-19 patients, despite the Italian Medicines Agency’s contrary recommendations [15]. Several studies have compared antibiotic prescribing before and after the start of the pandemic in outpatient settings [16,17,18,19,20,21,22,23,24,25,26]. These studies showed a decrease in antibiotic prescribing in outpatient settings during 2020, mostly due to measures implemented to prevent the spread of SARS-CoV-2, which resulted in a downtrend in the incidence of other respiratory tract infections. On the contrary, a study from Jordan analyzed the overall utilization of antibiotics (outpatient and hospital) and showed an increase in the use of certain antibiotics during the pandemic that are known to be associated with increasing resistance [27].

During the first months of the pandemic, an expert group, formed by the Ministry of Health and Social Welfare (Ministry) of the Republic of Srpska (an entity in Bosnia and Herzegovina), issued the first edition of the COVID-19 treatment protocol, in which the antibiotic azithromycin was recommended in addition to hydroxychloroquine/chloroquine in the treatment of COVID-19 pneumonia, but only for hospitalized patients [28]. According to clinical trials that did not show clinical benefits of the aforementioned medicines, azithromycin and hydroxychloroquine/chloroquine were excluded from the COVID-19 treatment protocol (third edition of the protocol) [29]. No other antibiotic was recommended as a special treatment option in the third and fourth editions of the COVID-19 treatment protocol [29,30], in which it was emphasized that all antibiotics should be prescribed in accordance with guidelines according to the principles of antimicrobial stewardship, with a note that antibiotics are not recommended to be prescribed for COVID-19 outpatients.

Most prescribed antibiotics come from the primary health care setting [31], and most often, irrational outpatient prescriptions of antibiotics are prescribed for patients with viral respiratory tract infections [32,33,34]. Therefore, it was of great importance to assess the use of antibiotics for COVID-19 patients in outpatient settings. The aims of this study were to analyze the utilization of antibiotics before (2018, 2019) and during the pandemic (2020) and the practice of prescribing antibiotics in outpatient settings (primary health care) for COVID-19 patients during the 2020–2022 period.

## 2. Results

### 2.1. Medicine Utilization

In the Republic of Srpska, the utilization of antibacterial medicines for systemic use (J01 group) was 25.54 DDD/1000 inhabitants/day (DIDs) in 2018, and 20.24 DIDs in 2019. During 2020, the utilization of the J01 group increased by 53.80% (31.13 DIDs) compared to 2019. In addition, the utilization of the J01 group comprised 99% of the total utilization of anti-infective medicines for systemic use (J group).

During all three observed years, the most commonly used antibacterial medicines for systemic use were beta-lactam antibiotics, J01C (Figure 1). In 2019, the utilization of beta-lactam antibiotics (J01C), other beta-lactam antibiotics (J01D), tetracyclines (J01A), and quinolones (J01M) decreased by 30.06%, 5.96%, 32.75%, and 8.07%, respectively, compared to 2018. Contrary, in 2020, the utilization of J01C, J01D, and J01M increased by 61.31%, 121.40%, and 8.49%, respectively, compared to 2019, while the utilization of J01A decreased by 2.31%. The utilization of macrolides, lincosamides, and streptogramins (J01F) increased by 3.72% in 2019 compared to 2018, while in 2020 it increased by 55.85% compared to 2019. It was observed that utilization decreased by 16.8% for sulfonamides and trimethoprim (J01E) and by 24.31% for aminoglycosides (J01G) in 2020 compared to 2019 (Table 1).

During the observational period, the most commonly used other beta-lactam antibiotics (J01D) were the first-generation cephalosporins (J01DB), whose utilization increased almost three times in 2020 compared to 2019 (Table 1).

During all three observed years, the most commonly used antibacterial medicine for systemic use, at the fifth level of the ATC classification, was amoxicillin (J01CA04), whose utilization increased by 100.08% in 2020 compared to 2019, when its utilization decreased by 40.43% compared to 2018. The utilization of amoxicillin/clavulanic acid (J01CR02) increased by only 8.32% in 2020 compared to 2019, while the utilization of cephalexin (J01DB01), cefixime (J01DD08), azithromycin (J01FA10), levofloxacin (J01MA12), and moxifloxacin (J01MA14) increased by 198.44%, 60.02%, 117.43%, 101.58%, and 124.60%, respectively. On the contrary, the utilization of doxycycline (J01AA02) decreased by 1.83% in 2020 compared to 2019 (Table 1).

### 2.2. Medical Records

During the observed period, the total number of COVID-19 outpatients registered in Web Medic with B34.2, U07.1, and U07.2 WHO International Classification Disease 10th revision (ICD-10) Codes was 16,565. The smallest number of COVID-19 outpatients was registered during the March–December 2020 (2020) period (2980). During the January–December 2021 (2021) and the January–June 2022 (H1 2022—first half of the year 2022) periods, there were 9307 and 4278 registered patients, respectively.

The total number of COVID-19 outpatients with the U07.2 WHO ICD-10 code was much higher than the total number of COVID-19 outpatients with the U07.1 WHO ICD-10 code during all three observed periods (Table 2). In addition, an increase in the share of the women in total number of COVID-19 outpatients was observed, as well as an increase in the share of COVID-19 outpatients younger than 15 years. Detailed sociodemographic characteristics of the analyzed patients are shown in Table 2.

At least one antibiotic was prescribed for 2713 (91.04%), 7729 (83.05%), and 3145 (73.52%) COVID-19 outpatients during 2020, 2021, and H1 2022, respectively. On a monthly basis, at least one antibiotic was prescribed for more than 55% of COVID-19 outpatients (except in April 2020, May 2020, July 2020, and July 2021) (Figure 2). The relation between the number of COVID-19 outpatients with WHO ICD-10 codes B34.2, U07.1, and U07.2, and the number of COVID-19 outpatients to whom was prescribed at least one antibiotic on a monthly basis is depicted on Figure 3.

The total number of antibiotic prescriptions was 3031 in 2020, 8988 in 2021, and 3343 in H1 2022. The three most commonly prescribed antibiotics during all three observed periods were doxycycline (15.07%), amoxicillin/clavulanic acid (20.35%), and azithromycin (43.34%) (Table 3). The share of azithromycin in the total number of prescribed antibiotics decreased by 27.85% in 2021 compared to 2020, while in H1 2022 it decreased by 2.68% compared to 2021. On the contrary, the share of amoxicillin/clavulanic acid in the total number of prescribed antibiotics shows an increase of 10.19% in 2021 compared to 2020. This trend continued in H1 2022 (by 12.9% compared to 2021). The share of doxycycline in the total number of prescribed antibiotics increased by 9.12% in 2021 compared to 2020, while in H1 2022 it remained almost unchanged compared to 2021.

A combination of two or three antibiotics was prescribed for 151 (5.07%) COVID-19 outpatients during 2020, for 313 (3.36%) COVID-19 outpatients during 2021, and for 34 (0.79%) COVID-19 outpatients during H1 2022. The most commonly prescribed combinations were doxycycline/amoxicillin/clavulanic acid and azithromycin/amoxicillin/clavulanic acid. Combinations were prescribed both on the first and control check-ups.

On the control check-up, antibiotic therapy was prolonged for 141 (4.73%) COVID-19 outpatients during 2020, for 746 (8.01%) patients during 2021, and for 138 (3.22%) patients during H1 2022.

## 3. Discussion

Studies that analyzed COVID-19’s impact on overall antibiotic prescribing trends in outpatient settings showed a decrease in antimicrobial prescribing rates during 2020 compared to the pre-pandemic period [16,17,18,19,20,21,22,23,24,25,26]. A different situation was observed in New York City, where an increase in antimicrobial prescribing rates was seen in outpatient settings from March to May 2020 compared to the same months in 2019 [35]. In that study, only 6% of documented antimicrobial indications referred to COVID-19. The London study, also conducted in outpatient settings, showed that 6158 of 33,708 SARS-CoV-2-positive patients received an antibiotic [16]; however, antibiotic indications were not analyzed. Our study is the first to date to focus on the prescribing of antibiotics for COVID-19 patients in outpatient settings.

Medicine repurposing is the process of identifying new indications for existing medicines and is considered an efficient and economical approach [36]. When no specific treatment is available and the development of a vaccine for a new disease is at its beginning, as it was for the new SARS-CoV-2 virus in 2020, the search for effective therapeutic agents is vital and urgent. At the beginning of the pandemic, azithromycin and hydroxychloroquine/chloroquine were repurposed, and both were part of the treatment of COVID-19 hospitalized patients in the Republic of Srpska. Although neither edition of the COVID-19 treatment protocol, issued by an expert group formed by the Ministry of the Republic of Srpska, recommended the use of antibiotics for COVID-19 outpatients, at least one antibiotic was prescribed for 91.04%, 83.05%, and 73.52% of COVID-19 outpatients in Banja Luka County during 2020, 2021, and H1 2022. All of those COVID-19 outpatients were prescribed an antibiotic on the first check-up. Although there was a downward trend in antibiotic prescribing during the study period, further investigation revealed that more than 55% of COVID-19 outpatients were prescribed an antibiotic on a monthly basis. In the Republic of Srpska, azithromycin was the most prescribed antibiotic in all three observed periods, probably related to the results of the study conducted in France [37]. Similar results were published in a study from Italy, where massive use of azithromycin in asymptomatic patients was observed [14]. Documented evidence showed that azithromycin was not effective in reducing the risk of COVID-19-related outcomes or the time to recovery compared to standard care alone [38,39]. Table 2 shows that the share of azithromycin among all prescribed antibiotics is decreasing in the Republic of Srpska, while the shares of amoxicillin/clavulanic acid and doxycycline are increasing. The decreasing trend of prescribed azithromycin is associated in a timely manner with the third and fourth editions of the COVID-19 treatment protocol issued by an expert group formed by Ministry of the Republic of Srpska [29,30], which were in line with WHO guidelines [7,8,9], but those protocols clearly noted the recommendation for azithromycin in hospital settings, not primary care settings. In 2020, due to its in vitro antiviral activity [40] and anti-inflammatory effects [41,42,43], doxycycline was proposed as a treatment for COVID-19 patients [44,45]. Afterwards, as a specific treatment for COVID-19, doxycycline was used in India [46] and Brazil [47], while in the United Kingdom it was used in the treatment of suspected COVID-19 pneumonia among patients with a high risk of adverse outcomes [48]. This practice was continued until 2021, when the PRINCIPLE trial showed no benefits from doxycycline treatment [49]. The increasing trend of doxycycline prescriptions in 2021 compared to 2020 in the Republic of Srpska (Table 2) might be in correlation with the above data, protocols from other countries, and the community-acquired pneumonia (CAP) protocol of the Ministry of the Republic of Srpska. That CAP protocol suggests tetracyclines as a second-line therapy for atypical CAP and macrolides as the first-line option [50]. In the same protocol, amoxicillin/clavulanic acid is the first choice for the treatment of typical CAP. On the other hand, our CAP protocol does not contain recommendations for the diagnosis and treatment of viral pneumonia as with similar protocols in other countries [35,51,52]. Our results revealed that more than 60% of COVID-19 outpatients were assigned the U07.2 WHO ICD-10 code, which means that the SARS-CoV-2 virus was not identified by the RT-PCR or antigen test among those patients. Since we were not able to link the medical records with the results of RT-PCR and antigen tests, we do not know if RT-PCR and antigen tests were negative or just not done. When SARS-CoV-2 and influenza viruses are co-circulating during the season, the Centers for Disease Prevention and Control recommends testing for both SARS-CoV-2 and influenza in outpatients with acute respiratory illness symptoms who do not require hospitalization [53]. The American Thoracic Society and Infectious Diseases Society of America suggest that antiviral and standard antibacterial treatment should be “initially prescribed for adults with clinical and radiographic evidence of CAP who test positive for influenza in the inpatient and outpatient settings” [54]. Vice versa, in the treatment of COVID-19 outpatients, antiviral therapy is recommended only “for patients who are at high risk of progressing to severe COVID-19” [54], while empirical antibiotics are not recommended for patients with mild, moderate, or severe COVID-19 [7,8,9].

Our study showed that at a control check-up, antibiotic therapy was more often prolonged in 2021 than in 2020, suggesting that during 2021 there were more patients with moderate COVID-19, but treated as outpatients. Thus, according to the COVID-19 treatment protocols in the Republic of Srpska, patients with moderate COVID-19 should be hospitalized [28,29]. The number of COVID-19 patients in 2021 (N = 9307) was three times higher than in 2020 (N = 2980). Such an increase in the number of patients was a potential overload for hospitals, and consequently, more patients had to be treated in outpatient settings. Until today, no recommendation was made to general practitioners in the Republic of Srpska for the treatment of moderate COVID-19 in outpatient settings, whereas the Italian Medicines Agency had already recommended specific antiviral treatment for specific subgroups of COVID-19 outpatients in 2020 [55]. Despite those recommendations, data from a recent Italian study revealed high antibiotic use for COVID-19 outpatients [15], but at a much smaller amount than in our study.

A set of activities to enhance rational antibiotic prescribing had been running continuously in the Republic of Srpska before the pandemic. Bojanić at al. showed that during the 2010–2015 period, all of these activities resulted in a decrease in antibiotic utilization (from 15.6 to 18.4 DIDs for the J01 group) and lower use of antibiotics compared to neighboring countries [56]. However, our study revealed that outpatient antibiotic utilization increased in the Republic of Srpska after the Bojanićs’ study (25.54 DIDs in 2018; 20.24 DIDs in 2019 for the J01 group). Data on antimicrobial consumption by the European Centre for Disease Prevention and Control shows that in most European countries, outpatient antibiotic utilization decreased in 2020 compared to 2019 [31]. In the Republic of Srpska, outpatient antibiotic utilization, on the contrary, increased by 53.80% (31.13 DIDs) in 2020, mostly due to increase in the utilization of amoxicillin (J01CA04), cephalexin (J01DB01), cefixime (J01DD08), azithromycin (J01FA10), levofloxacin (J01MA12), and moxifloxacin (J01MA14). It is not allowed by law to dispense an antibiotic without a prescription in the Republic of Srpska. Fines for violating the law are well defined, and nowadays, the practice of dispensing an antibiotic without a prescription is reduced to a minimum [57,58]. In order to overcome and prevent serious consequences caused by potentially over-prescribed antibiotics, it is crucial that practices, healthcare professionals, and policy makers work together in promoting and evaluating a set of activities which have been started more than decade ago.

Thus, further research is needed to determine the potential causes of the increase in antibiotic utilization during the COVID-19 pandemic.

## 4. Strength and Limitations

Our study has several strengths. Antibiotic prescribing rates are analyzed specifically in the COVID-19 outpatient population. The study was conducted during the March 2020–June 2022 period; therefore, it covered both the lockdown period and the period without strict anti-pandemic measures. However, our study also has several limitations. Clinical manifestations, and laboratory and imaging results of COVID-19 outpatients were not analyzed in our study. Therefore, ICD codes may not match the real clinical indication. The rate of co-infections for COVID-19 outpatients was not analyzed. The results of medical records’ data refer to 16,565 COVID-19 outpatients registered in Banja Luka County, a region of the Republic of Srpska. Therefore, those results may not be representative of the whole entity.

## 5. Materials and Methods

This was a retrospective observational study of outpatient antibiotic utilization (focused on the COVID-19 pandemic) based on medicine utilization data obtained from the Public Health Institute (PHI) of the Republic of Srpska and medical records data obtained from the Primary Healthcare Center (PHC) of the Banja Luka County.

### 5.1. Medicine Utilization

Every year, PHI of the Republic of Srpska publishes a document with the total medicine utilization in the Republic of Srpska for the past year. Those data refer both to hospital and outpatients’ (out-of-hospital) use of medicines, which are presented separately. The same report does not show data separately by municipalities and cities. Data on outpatients’ use of medicines refer to all medicines dispensed from 466 pharmacies in the Republic of Srpska [59]. Our study analyzed the data on outpatients’ use of antibacterial medicine for systemic use (J01 group) before (2018, 2019) and during the first year of the pandemic (2020).

Medicine utilization analysis of the PHI data was undertaken using the ATC (Anatomical Therapeutic Chemical classification)/DDD (Defined Daily Dose) methodology [60], which is the internationally accepted methodology for measuring medicine utilization within and across populations [56,61,62,63,64]. The DDD value is not a recommended therapeutic dose for a specific indication, but is determined based on the average therapeutic dose used for the basic indication. It is a suitable measure to describe and compare medicine utilization during years. Data on outpatient antibiotic utilization are expressed in DDD/1000 inhabitants/day (DIDs) for comparative purposes [60,62,63]. The growth index (GI) was used to determine the change in medicine utilization with regard to the previous year. It was calculated using the following formulae:$$\frac{\mathrm{DIDs}\;\left(2020\right)-\mathrm{DIDs}\;\left(2019\right)}{\mathrm{DIDs}\;\left(2019\right)}\cdot 100$$
to determine the change in medicine utilization in 2020 versus 2019, and
$$\frac{\mathrm{DIDs}\;\left(2019\right)-\mathrm{DIDs}\;\left(2018\right)}{\mathrm{DIDs}\;\left(2018\right)}\cdot 100$$
to determine the change in 2019 versus 2018. The GI value is presented as percentages.

### 5.2. Medical Records

Data were obtained from the Banja Luka County PHC center. Banja Luka is the biggest city in the Republic of Srpska, where the PHC center delivers health care to 250,000 patients and has specialized units for COVID-19 outpatients. The study included all COVID-19 outpatients registered in the Web Medic database, an electronic platform of the primary health care system that was in use from 2009 to 2022. If a patient was registered in Web Medic with B34.2, U07.1, and U07.2 WHO ICD-10 Codes, he was considered as a COVID-19 outpatient. WHO ICD-10 codes U07.1 and U07.2 are Emergency Codes that were activated by WHO in February 2020 in response to a pandemic. WHO ICD-10 code U07.1 refers to COVID-19 with identified SARS-CoV-2 virus, and U07.2 also refers to COVID-19, but without identified SARS-CoV-2 virus [65], while B34.2 is the WHO ICD-10 code for unspecified coronavirus infections [66]. In the PHC center of Banja Luka County, WHO ICD-10 Emergency Codes have been in use since the end of September 2020. Until then, only the B34.2 WHO ICD-10 code was used for COVID-19. Since the aim of this study was to analyze the use of antibiotics for COVID-19 treatment, patients to whom antibiotics were not prescribed on the first check-up were not included in the study due to the assumption that those patients developed secondary bacterial infections. Data were analyzed during three periods: March–December 2020 (2020), January–December 2021 (2021), and January–June 2022 (H1 2022—first half of the year 2022). Using these criteria, 16,565 medical records were analyzed.

The Medical Faculty of the University of Banja Luka has sent a formal request to the PHC for the sharing of epidemiological data for the purpose of this research. PHC sent a confidential email about COVID-19 to the main researchers (A.G.J. and N.B.) in the form of data sheets for each of the three observed periods (2020, 2021, and H1 2022), without the patients’ names or any detail that could in any way identify patients’ names. The COVID-19 data sheets comprised the following patients’ data as researchers requested: (1) the identifying number of the patient from the Web Medic medical records (ID); (2) gender; (3) age; (4) date of birth; (5) date of the patient’s visit to the PHC; (6) temporary COVID status; (7) WHO ICD-10 code; (8) date of the prescription prescribed to the patient at that visit; (9) medicine(s) prescribed to the patient at that visit (manufacturer name, International Nonproprietary Name (INN), and dose regimen).

Data extractions were undertaken manually in Excel sheets (available on request) by two researchers independently to ensure the reliability of the information. First researcher (N.B.) made extraction, and second researcher (A.G.J.) double-checked the data extraction to ensure the accuracy of the extracted information [67,68].

Data analysis for all three observed periods included the following: (1) the percentage of COVID-19 outpatients to whom at least one antibiotic was prescribed; (2) the total number of antibiotic prescriptions; and (3) the share of individual prescribed antibiotics (ATC level V) in the total number of prescribed antibiotics. The percentage of COVID-19 outpatients to whom at least one antibiotic was prescribed was also analyzed on a monthly basis.

In addition, the analysis included the percentage of COVID-19 outpatients to whom an antibiotic combination was prescribed and the percentage of COVID-19 outpatients to whom antibiotic therapy was prolonged on the control check-up. If more than one antibiotic was prescribed to one patient on the same visit to the GP, it was considered that an antibiotic combination was prescribed, while, if more than one antibiotic was prescribed to one patient in two or more visits, it was considered that antibiotic therapy was prolonged on the control check-up.

The results are presented as counts, percentages, and trend analyses.

No specific ethical approval was sought as only aggregated, anonymized data was used for analyses, with Medicine Faculty personnel involved in the study. This is similar to other study of this nature [56,61,64,69].

## 6. Conclusions

Results of our study revealed that on a monthly basis, at least one antibiotic was prescribed for more than 55% of COVID-19 outpatients registered in Banja Luka County. The three most common prescribed antibiotics for COVID-19 outpatients were azithromycin, amoxicillin/clavulanic acid, and doxycycline. Contrary to other European countries, in the Republic of Srpska, outpatient antibiotic utilization increased by 53.80% in 2020 compared to 2019. The trend of repurposing antibiotics with antiviral and anti-inflammatory activities for COVID-19 treatment might be a double-edged sword. The long-term effect of this practice might be an increase in AMR and a loss of antibiotic effectiveness. According to the change in the etiology of respiratory tract infections, there is an urgent need to update the existing national treatment guidelines worldwide. There is an obvious lack of studies in the field of antibiotic prescribing for COVID-19 outpatients. Since the majority of antibiotics are prescribed in outpatient settings [28] and most often irrationally for viral respiratory tract infections [29,30,31], the regulation of the treatment for COVID-19 outpatients should be considered an important part of antimicrobial stewardship today. Many antiviral medicines were analyzed in the past two years for COVID-19 outpatients. Practices, healthcare professionals, and policy makers should work together to consider the implementation of these medicines in treatment guidelines in order to provide wider treatment options for physicians in primary health care settings.

## Figures and Tables

**Figure 1 antibiotics-12-00308-f001:**
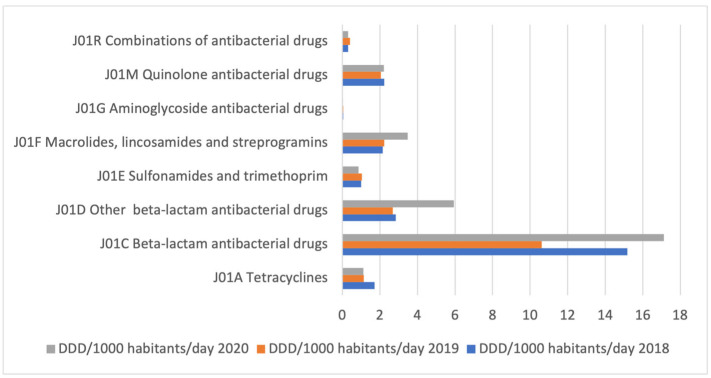
The utilization of antibacterial medicines for systemic use (J01 group) during the 2018–2020 period in the Republic of Srpska, at the third level of ATC classification, expressed in DDD/1000 inhabitants/day (DIDs).

**Figure 2 antibiotics-12-00308-f002:**
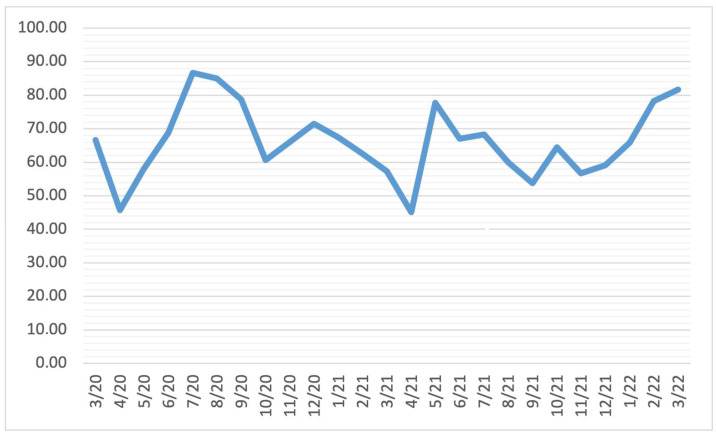
The percentage of antibiotic prescriptions given to COVID-19 outpatients with WHO ICD-10 codes B34.2, U07.1, and U07.2 during the March 2020–June 2022 period, presented monthly.

**Figure 3 antibiotics-12-00308-f003:**
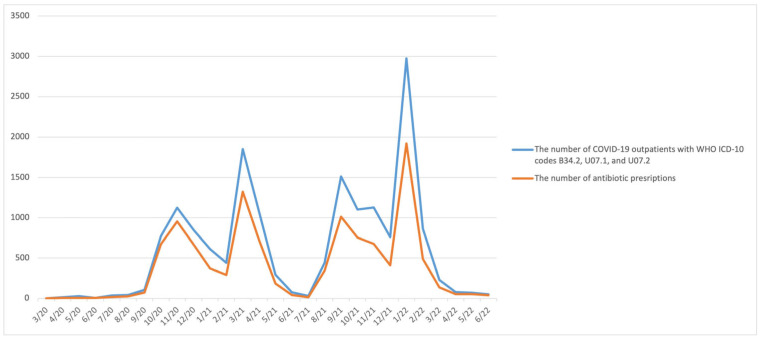
The relationship between the number of COVID-19 outpatients with B34.2, U07.1, and U07.2 WHO ICD-10 codes and the number of antibiotic prescriptions for COVID-19 outpatients during the March 2020–June 2022 period, presented monthly.

**Table 1 antibiotics-12-00308-t001:** The outpatient utilization of antibacterial drugs for systemic use (ATC group J01), expressed in DIDs with growth index (GI).

	2018	2019	GI 19/18 *	2020	GI 20/19 **
III level of the ATC classification					
J01C (beta-lactam antibiotics)	15.17	10.61	−30.06%	17.12	61.31%
J01D (other beta-lactam antibiotics)	2.85	2.68	−5.96%	5.93	121.40%
J01F (macrolides, lincosamides, and streptogramines)	2.15	2.23	3.72%	3.48	55.85%
J01M (quinolones)	2.23	2.05	−8.07%	2.22	8.49%
J01A (tetracyclines)	1.71	1.15	−32.75%	1.13	−2.31%
J01E (sulfonamides and trimethoprim)	1.01	1.04	2.97%	0.86	−16.80%
J01X (other antibiotics)		0.43		0.36	−17.13%
J01G (aminoglycosides)	0.05	0.05	0.00%	0.04	−24.31%
IV level of the ATC classification					
J01DB (first-generation cephalosporins)	1.84	1.59	−13.59%	4.77	200%
J01DC (second-generation cephalosporins)	0.72	0.61	−15.28%	0.52	−14.75%
J01DD (third-generation cephalosporins)	0.29	0.46	58.62%	0.63	36.96%
J01DE (fourth-generation cephalosporins	0	0.000178		1.81 × 10^−5^	−89.83%
J01FA (macrolides)	2.06	2.14	3.88%	3.41	59.34%
J01FF (lincosamides)	0.08	0.09	12.50%	0.07	−22.22%
J01MA (fluoroquinolones)	2.11	1.99	−5.69%	2.22	11.56%
J01MB (other quinolones)	0.13	0.05	−61.54%	4.82 × 10^−5^	−99.90%
V level of the ATC classification					
J01CA04 (amoxicillin)	11.08	6.6	−40.43%	13.21	100.08%
J01DB01 (cephalexin)	1.84	1.6	−13.04%	4.77	198.44%
J01CR02 (amoxicillin/clavulanic acid)	3.04	2.99	−1.64%	3.24	8.32%
J01FA10 (azithromycin)	1.14	1.28	12.28%	2.79	117.43%
J01MA02 (ciprofloxacin)	1.52	1.45	−4.60%	1.33	−8.76%
J01AA02 (doxycycline)	1.45	1.15	−20.69%	1.13	−1.83%
J01EE01 (sulfamethoxazole, trimethoprim)	1.01	1.04	2.97%	0.86	−16.80%
J01MA12 (levofloxacin)	0.33	0.32	−3.03%	0.65	101.58%
J01CE02 (phenoxymethylpenicillin)	0.9	0.84	−6.67%	0.55	−34.31%
J01DC02 (cefuroxime)	0.68	0.58	−14.71%	0.51	−12.90%
J01FA09 (clarithromycin)	0.49	0.49	0.00%	0.41	−17.35%
J01DD08 (cefixime)	0.19	0.24	26.31%	0.38	60.02%
J01DD04 (ceftriaxone)	0.07	0.18	157.14%	0.2	9.19%
J01FA01 (erythromycin)	0.42	0.34	−19.05%	0.19	−42.56%
J01MA06 (norfloxacin)	0.21	0.19	−9.52%	0.17	−11.51%
J01CA01 (ampicillin)	0.12	0.15	25.00%	0.09	−36.97%
J01MA14 (moxifloxacin)	0.03	0.04	33.33%	0.08	124.60%
J01DD13 (cefpodoxime)	0.02	0.04	100.00%	0.05	21.39%

* GI 19/18—growth index that represents the change in antibiotic utilization in 2019 versus 2018. ** GI 20/19—growth index that represents the change in antibiotic utilization in 2020 versus 2019.

**Table 2 antibiotics-12-00308-t002:** Socio-demographic characteristics and distribution of COVID-19 outpatients according to diagnosis.

Variables	2020		2021		H1 2022 ^1^	
	N (%)		N (%)		N (%)	
Gender						
Female	1559 (52.32)		5217 (56.06)		2508 (58.63)	
Age (years)						
<15	15 (0.50)		164 (1.76)		131 (3.06)	
16–25	257 (8.62)		891 (9.57)		347 (8.11)	
26–35	463 (15.54)		1320 (14.18)		607 (14.19)	
36–45	748 (25.10)		1948 (20.93)		904 (21.13)	
46–55	547 (18.36)		1581 (16.99)		691 (16.15)	
56–65	514 (17.25)		1704 (18.31)		757 (17.70)	
>66	436 (14.63)		1698 (18.24)		840 (19.64)	
Number of patients according to WHO ICD-10 ^2^ Codes						
B34.2 ^3^	229 (7.68)		327 (3.51)		212 (4.96)	
U07.1 ^4^	645 (21.64)		3236 (34.77)		1399 (32.70)	
U07.2 ^5^	2106 (70.67)		5744 (61.72)		2667 (62.34)	

^1^ Data from January–June 2022 period, first half of the year 2022. ^2^ World Health Organization International Classification of Disease 10th revision. ^3^ Unspecified coronavirus infections. ^4^ Coronavirus disease 2019 (COVID-19), virus identified. ^5^ COVID-19, virus not identified.

**Table 3 antibiotics-12-00308-t003:** The share of prescribed antibiotics expressed as a number (%) of total prescriptions, presented yearly.

	N (%)		
Antibiotics According to V Level of ATC Classification	2020	2021	H1 2022
amoxicillin	38 (1.25)	145 (1.61)	84 (2.51)
amoxicillin/clavulanic acid	284 (9.37)	1758 (19.56)	1085 (32.46)
azithromycin	2009 (66.28)	3454 (38.43)	1195 (35.75)
cefixime	48 (1.58)	187 (2.08)	48 (1.44)
ceftriaxone	5 (0.16)	45 (0.5)	
cefuroxime	5 (0.16)	16 (0.18)	4 (0.12)
ciprofloxacin	30 (0.99)	133 (1.48)	46 (0.12)
doxycycline	236 (7.79)	1520 (16.91)	559 (16.72)
levofloxacin	310 (10.23)	400 (4.45)	161 (4.82)
moxifloxacin	41 (1.35)	1181 (13.14)	109 (3.26)

## Data Availability

Available on request.

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
