# Peer review of "Antibiotic Utilization during COVID-19: Are We Over-Prescribing?"

_antibiotics, 2023, doi:10.3390/antibiotics12020308_

Round 1
Reviewer 1 Report
Antibiotics and COVID-19: Are we leading to a post-antibiotic era? by Bednarčuk et al. is an intersting and timely paper. However, I have some suggestions:
1. Please improve the quality of the figures 1,2 and 3.
2. There are few studies in the literature looking at drugs of abuse trend and antimicrobials in the United States. One paper has both wastewater and prescriptions date. Do you guys have similar conclusion? If so, why? If not why not? These may be different domain but yet an interesting observation for science.
3. Change the title of the paper. This title is broad and does not justify the work of your nation.
Author Response
Thank you for your contribution in reviewing our manuscript.
We tried to answer point by point at your comments and suggestions. Please see the attachment.
Your comment were very useful.
Best regards

Reviewer 2 Report
Dear Authors,
This article contains significant findings that are worth publication. However, I have minor comments that need to be addressed before this article is published.
Please refer to the comment in the attached file.
Kind regards,

Author Response
Respected reviewer,
we found your comments and suggestions as very useful. We believe that our manuscript is strengthener and clearer than before.
You may find our response point by point in the attachment.
Best regards
